# Tooth Abnormalities and Their Age-Dependent Occurrence in Leukemia Survivors

**DOI:** 10.3390/cancers15225420

**Published:** 2023-11-15

**Authors:** Anna Jodłowska, Lidia Postek-Stefańska

**Affiliations:** Department of Pediatric Dentistry, Faculty of Medical Sciences in Zabrze, Medical University of Silesia, 40-055 Katowice, Poland; lstefanska@sum.edu.pl

**Keywords:** tooth abnormalities, acute lymphoblastic leukemia, dental development, chemotherapy

## Abstract

**Simple Summary:**

Despite the multidrug nature of anticancer treatment, attempts are still being made to determine the relationship between the duration of chemotherapy, the dose of individual drugs, and the occurrence of dental developmental abnormalities. There are relatively few papers devoted to this issue, and all of them are based on the study groups composed of individuals receiving various treatment regimens. The current study includes a group of acute lymphoblastic leukemia survivors who underwent chemotherapy according to the ALL IC-BFM 2002 protocol. Contrary to the observations of some authors, the results of the current research suggest that the age at the start of chemotherapy is likely to be the strongest risk factor for toxic injury during tooth development. A small study cohort is a major limitation, but an evaluation of similar relationships at larger research centers would be helpful in better understanding the problem.

**Abstract:**

The multidrug nature of anticancer treatment and different treatment protocols used in the studies are likely to be a major limitation in establishing real risk factors determining the occurrence of dental abnormalities. The authors aimed to establish a relationship between the duration and the dose of chemotherapy and the number of tooth adverse effects in the group receiving the same treatment. Of the 40 anticancer therapy recipients who attended the outpatient dental clinic, 7 leukemia survivors receiving the treatment according to the ALL IC-BFM 2002 protocol were selected. The study group consisted of four females and three males aged 92 to 207 months at the time of dental examination and 29 to 91 months at leukemia diagnosis. As a result of the clinical and radiological examination, dental abnormalities such as agenesis, tooth size reduction, root abnormalities, and taurodontia were identified, and the medical records of all survivors were reviewed in terms of drugs administered, their doses, and treatment schedules. No correlation was observed between the treatment duration of an intensive therapy, the entire therapy, and the number of tooth abnormalities. No relationship was also found between the number of dental abnormalities and the cumulative dose of vincristine, L-asparaginase, methotrexate, cyclophosphamide, cytarabine, and 6-mercaptopurine. The age at the onset of antineoplastic therapy is likely to be the strongest risk factor for toxic injury during tooth development.

## 1. Introduction

It has been well documented that exposure to chemotherapy always causes side effects in the human body [1,2,3]. Their occurrence depends on a number of factors, sometimes unknown. There is strong evidence that dental abnormalities are late adverse effects of antineoplastic therapy when it is introduced at a young age [4,5,6,7,8,9,10,11,12,13,14]. When the abnormalities concerning developing structures are discussed, age is frequently considered an important determining factor. It has already been proven that young cancer survivors presenting with the most severe dental changes had been receiving chemotherapy before they were 3–5 years old. Furthermore, they exhibited no dental abnormalities if they had been treated at the age between 43 and 61 months [5,6,10,15,16,17,18,19,20,21,22]. Some scientific works investigated the influence of individual therapeutic agents on the frequency of dental anomalies appearance [5,15,16,23]. However, the multidrug character of anticancer treatment seems to be a major limitation in terms of reliable assessment of the above-mentioned dependencies. Although the multiagent concept of chemotherapy minimizes its toxicity, it has already been noticed that even a small drug dose can affect the developing structure [15]. Moreover, the non-homogenous groups of survivors were predominantly examined so far. A study conducted on a homogenous cohort treated according to the same treatment protocol could bring a new perspective on the problem. In the literature, neuroblastoma, nephroblastoma, non-Hodgkin’s lymphoma, or leukemia survivors were screened when it comes to disturbed dental development [8,9,10,11,12,13,14,24]. No analyses involving drug doses’ impact on the occurrence of dental abnormalities were investigated. That is why the drug dose- or the age dependency of dental abnormalities’ occurrence remains in question.

Leukemia is the most common childhood cancer, and its incidence peaks between 1 and 5 years [1,25,26]. As many as 70–90% of young patients developing diagnosed acute lymphoblastic leukemia (ALL) survived for a long time over the past few decades [1,25,27]. The pediatric precursor B-cell ALL (pre-B ALL) is currently treated according to two risk-dependent regimens. ALL risk stratification is generally based on two prognostic factors: initial white blood cell (WBC) count and the patient’s age at diagnosis [1,28,29,30,31]. According to the National Cancer Institute criteria, pre-B ALL is classified into standard- (SR) and high-risk categories (HR) [1,28,32]. For the SR-ALL, a 35-day induction with lumbar puncture (l.p.) plus intrathecal (i.th.), intravenous (i.v.), and oral (p.o.) chemotherapy is recommended. Vincristine (VCR), glucocorticoids (e.g., prednisone and dexamethasone), daunorubicin (DR), and L-asparaginase (LASP) are generally used in the course of the intensification period [1,32]. This protocol is followed by 28-day consolidation, including the same ways of drug administration with slightly changed anticancer drugs. When it comes to HR disease, the consolidation protocol is longer (56 days), and this is enriched with the administration of two additional anticancer drugs [28,33]. In the literature, the next post-remission phase of anticancer treatment is sometimes described as delayed intensification. These consolidation/intensification regimens are frequently based on the systemic and intrathecal administration of corticosteroids, VCR, LASP, doxorubicin (DXR), Ara-C, 6-Mercaptopurine (6-MP), cyclophosphamide (CP), and high-dose MTX [1,32,34]. Although intensive treatment is usually effective, long-lasting maintenance therapy with orally administered MTX and 6-MP is always recommended. In patients at a very high risk of relapse, even though remission is observed, an allogeneic or autologous stem cell transplantation (SCT) is introduced, followed by total body irradiation (TBI) [1,29]. More recently, immune-directed therapy has been used in HR-relapsed ALL patients [1,35]. Regardless of the type of therapy, leukemia regimens are the longest of all cancer treatment protocols.

Since the year 2002, leukemia has been treated according to three risk-dependent protocols based on pioneering findings of the International Berlin–Frankfurt–Munster Study Group concerning the early response to the therapy of prednisone on day 8 and percentage of the number of bone marrow blasts on day 15 (ALL IC-BFM 2002). Standard-(SR), intermediate-(IR), and high-risk ALL (HR) were diagnosed [33,36,37]. The treatment has generally begun with a 62-day intensive regimen referred to as induction (Protocol I), including seven or eight antileukemic drugs, and was followed by 2-month consolidation (Protocol M) with the use of 6-MP and MTX. Delayed intensification (Protocol II) was then realized during the next 50 days and was based on the administration of the drugs used for induction. After intensive treatment had been completed, the 75-week maintenance therapy with weekly MTX and daily 6-MP oral administration was introduced. A detailed description of all protocols with drug doses has been included in the paper of Stary et al. [36].

Less aggressive regimens used at the beginning of the leukemia therapy caused lower survival rates. Intensification of the treatment significantly improved event-free survival rates, especially in terms of the possible development of subclinical meningeal leukemia [27,33,38]. Then, it was proven that more intensive maintenance treatment also improves survival rate compared to an extra year or longer low-dose maintenance therapy [39]. However, the maintenance therapy must also be long enough to lower the risk of relapse despite the higher risk of death in remission [1,3,39]. Young patients treated for ALL need to be monitored in the course of intensive therapy because delayed recognition of the intensive therapy toxicity can lead to unpredictable acute side effects, including mortality [27]. Moreover, the patients treated for ALL may present with long-term adverse effects, such as retarded growth, thyroid dysfunction, infertility, second cancer, and neurologic or learning disability. Some patients may not remain free of relapse. Tooth abnormalities have also been observed to be late adverse effects in leukemia survivors; however, the relationship between the duration of the therapy and the prevalence of dental anomalies was not found [22].

The objective of the study was to establish a relationship between the duration of treatment, the dose of chemotherapy, and the number of tooth abnormalities in pre-B ALL survivors receiving the treatment according to the same protocol.

## 2. Materials and Methods

A group of 40 cancer survivors attended the Outpatient Dental Clinic of Academic Centre of Dentistry and Specialized Medicine in Bytom, Poland, following the inclusion criteria established in the previous cross-sectional study [15]: aged 5–18 years and a minimum 2 years after anticancer therapy completion. The patients were in remission, and they were monitored regularly in their Outpatient Hematology and Oncology Clinic of Upper Silesian Children’s Health Center in Katowice, Poland. An attempt to identify pediatric leukemia survivors was made. Of the eight leukemia survivors, seven participants diagnosed with pre-B ALL code 204.0 according to the International Classification of Diseases (ICD-9) were selected. They were treated under similar conditions in the same hospital between 2005 and 2013. The anticancer therapy was carried out according to similar protocols depending on the risk of malignancy. Five of the seven individuals were classified as having SR-ALL, and two patients presented with IR disease. The ALL IC-BFM 2002 protocol was administered to all of them with small variations depending on the risk diagnosed. The medical records of all survivors were thoroughly reviewed in terms of drugs administered, their doses, and treatment schedules. All data are shown in Table 1 and Table 2. At the moment of enrolling pre-B ALL patients, we observed a 24-month event-free survival minimum and a 78-month event-free survival maximum after the maintenance therapy completion in six selected individuals. One participant developed two bone marrow relapses after he had 18-month and 14.5-month event-free periods, respectively. The new protocols were introduced to him (REZ BFM 2002 and INT ReAll 2010, respectively), but the treatment did not overlap with the critical developmental period of his dentition. This is why the patient was not excluded from the study, and both additional regimens were not undertaken in comparative analysis. The relapsed survivor received prophylactic central nervous system radiotherapy (CSN RT) when aged 12 and stem cell transplantation (SCT) following the last course of chemotherapy at the age of 17. Neither RT nor SCT was given to the remaining study participants.

ALL cancer survivors, after caregivers gave their written consent for participation in the study, underwent a dental examination, including a panoramic radiograph. Thorough analysis revealed a total of 37 teeth with different developmental anomalies, and one tooth failed to erupt (Figure 1). Detailed information is summarized in Table 1. 

In order to establish a relationship between treatment duration and the number of dental abnormalities, the duration of a real period during which the anticancer drugs were taken by patients was calculated. For this purpose, the duration of intensive therapy after excluding all the treatment breaks was determined. Additionally, the duration of the entire therapy was also taken into correlation analysis (Table 3). In the next step, the cumulative doses of the drugs that were administered to all study participants were calculated. VCR (i.v.), LASP (i.v.), MTX (i.th.), CP (p.i.), Ara-C (i.v.), and 6-MP (p.o.) were chosen for analysis. Subsequently, an attempt was made to assess the effect of the drug dose size on the occurrence of tooth abnormalities (Table 4).

### Statistical Analysis

For treatment duration, the median and standard deviation were calculated to assess the similarity of protocols within the group (Table 1). A relationship between the treatment duration of intensive therapy (breaks excluded), the treatment duration of the entire therapy (breaks included), and the number of abnormalities was established with the use of Kendall’s tau-b nonparametric rank test (Table 3). Kendall’s tau-b coefficient was also used to assess a correlation between the number of abnormalities and the cumulative dose of these drugs that were administered to all study participants (Table 4). All calculations were performed using the statistical program SPSS, version 28.3.

## 3. Results

No statistically significant correlation was observed between the treatment duration of intensive therapy (breaks excluded), the treatment duration of the entire therapy (breaks included), and the number of tooth abnormalities (tau-b = −0.12; *p* = 0.732 for intensive therapy; tau-b = 0.21; *p* = 0.534 for the entire therapy) (Table 3).

No statistically significant relationship was found between the number of dental abnormalities and the cumulative dose of VCR, LASP, MTX, CP, Cyta, and 6-MP (*p* > 0.050 for all pairs of variables) (Table 4).

## 4. Discussion

Cross-sectional studies of large groups of patients undergoing chemotherapy were usually difficult due to the treatment protocols changing over the years [27]. Therefore, the determination of cause and effect relationship in terms of both acute and long-term adverse treatment sequelae met some limitations. Although dental developmental abnormalities as the anticancer treatment effects have been widely documented, the type of therapy was usually not analyzed in the past [4,7,19,20,22,40,41]. Many risk factors have been considered, but the age at the onset of the therapy was the most frequently quoted [4,5,6,10,17,18,19,20,21,22,42]. Cubukcu et al. emphasize that the same treatment protocol may cause different changes in different teeth depending on the stage of development [7]. Although the dental developmental stage, thereby the patient’s age at the start of the anticancer therapy, seems to be the strongest risk factor, the impact of treatment duration and particular toxic agents has been analyzed in some recent papers [5,15,16,23,42,43,44]. Proc et al. confirmed that chemotherapy has a significant impact on odontogenesis, but an established number of tooth abnormalities did not differ with treatment duration. The influence of the drugs and their doses has not been taken into study consideration [5]. Krasuska-Sławińska et al. revealed positive correlations between hypodontia and age at the onset of treatment and the use of VCR, DXR, CP, ifosfamide (IF), and VP-16 using Spearman’s rho. The number of missing teeth increased with increasing drug dose and treatment duration. They reported a positive correlation between microdontia and treatment duration and the use and doses of VCR, DXR, CP, IF, and VP-16, and between taurodontia and age at treatment onset and VCR administration and its dose. The same study established positive correlations between dental root resorption and age at diagnosis and therapy with VCR, cisplatin (CDDP), CP, IF, VP-16, and DXR. The prevalence of enamel defects was positively correlated with age at the start of treatment and the use of VCR, platinum agents, CP, DXR, IF, and VP-16. Krasuska-Sławińska et al. generally observed positive dose-dependency between the administration of analyzed drugs and the occurrence of particular tooth abnormalities excluding hypoplasia in carboplatin (CBDCA) recipients [23]. In the study, the only information on drug dose seemed to be a single one because the dosing method, such as mg/m^2,^ has been supplemented. Despite the fact that doses presented in the study are rather likely to be cumulative, they both usually increase with age. Based on the data revealed in the literature, the correlation could not be positive over the entire developmental period. However, the number of abnormalities in particular age groups was not taken into study consideration. It was only proven that tooth disturbances were more prevalent in patients after chemotherapy than in controls. Kaste et al. also examined survivors with different cancer diagnoses and different multidrug treatment regimens administered. They analyzed alkylating agents, the most widely used anticancer drugs in pediatric oncology, in leukemia treatment as well. They reported a dose-dependent relationship between the exposition to alkylating agents and increased risk for dental sequelae, especially in survivors diagnosed before 5 years of age. However, the study was based on a self-reported questionnaire, which seems to be a strong limitation [43]. Stolze et al. also revealed a dose-dependent increased risk for dental abnormalities after the administration of alkylating agents. However, the authors indicate a major limitation of the study, as the historical data were obtained from the dentists without the possibility of X-ray analysis [42]. In another study analyzing CP administration, a dose-dependent impact on Holtta’s defect index (HDI) was established with an increase of 13 HDI score after receiving a dose higher than 7.5 mg/m^2^ in relation to patients not being treated with CP. Although ALL and Wilms tumors were the most frequently diagnosed conditions in the study participants, the impact of many concomitant factors cannot be excluded [44].

These observations cannot be confirmed based on previously published studies on homogenous cohorts [11,15,16,21]. The authors of the current study analyzed VCR, DXR, CP, VP-16, carboplatin (CBDCA), and actinomycin D (ACTD) administration in their previous study due to their most frequent use in the examined participants and the range between 73,33% minimum and 85,71% maximum of individual drugs’ recipients with teeth affected was noted. There were no statistically significant differences in the number of abnormal teeth between the analyzed drug groups when it comes to treatment duration and mean cumulative drug dose administered, no matter whether an intensive or entire therapy was taken into consideration [15]. However, all of the above-described clinical studies had their participants treated according to different protocols. Determining the drug dose impact and treatment duration toxic effect of an individual drug is not possible until the administration of every anticancer agent overlaps with the entire therapy.

As was already mentioned in the recent studies, the authors of the present study revealed no correlation between the number of dental abnormalities and either treatment duration or drug dose. It was also concluded that even the lowest drug dose could damage immature tissues. So common in the literature, the method of evaluating the toxicity of a single therapeutic agent given in the multidrug protocol was considered a strong limitation [15,16]. The influence of other factors such as age at diagnosis, treatment duration, and different protocols used are also important. An analysis of a more homogenous group of survivors seems to be more reliable. Wilberg et al. also met some limitations in their study and saw the need to assess a large sample size with a single diagnosis, relatively homogenous therapy, and long observation time [17]. Some authors have examined pediatric cohorts with the same cancer diagnosis, but they reported the prevalence of dental abnormalities having no interest in terms of treatment-influencing factors [12,13,14,24,45,46,47]. Rosenberg et al. studied 17 long-term survivors of childhood ALL for altered dental root development, and no treatment analysis was conducted [14]. In another study, 52 ALL and acute myeloid leukemia (AML) survivors treated according to the same protocol referred to as AIEOP (Italian Association of Paediatric Hematoncology) were controlled in order to perform dental sequelae assessment. No treatment details were taken into consideration [24]. Kinirons et al. examined a dental caries experience in relation to the chemotherapy duration, and they found no significant differences in the number of carious lesions in patients in remission of ALL. There were no dental abnormalities observed in the study [47]. Cordova Maciel et al. observed at least one dental abnormality in 80.4% of patients treated for ALL. Participants younger than 5 years were significantly more affected. There was no investigation on the relationship between treatment conditions and dental abnormalities occurrence in the study [45]. There is little research in the literature analyzing in detail the effect of anticancer treatment on the disturbance of tooth development based on cohorts treated according to the same protocol [11,21]. Marec-Berard et al. conducted a dental examination on 27 nephroblastoma survivors treated between 1994 and 1998. They reported that therapy duration and drug dose do not contribute to the occurrence of dental abnormalities. Children were treated with VCR, ACTD ± DXR, but there are no statistical results relating to the treatment duration and individual drug doses in the paper [11]. An interesting analysis was conducted by Minicucci et al. on leukemia survivors treated from 1980 to 1990 according to four different protocols: three Brazilian Cooperative Protocols (ALL 1980, 1982, and 1985) and the BFM86 Protocol. Three groups were created: GI with HR disease and cranial radiotherapy administered, GII with LR disease, and GIII based on the BFM68 Protocol. The number of patients with abnormal teeth diagnosed was higher than the number of survivors free of anomalies in every group. The statistically significant differences between participants with or without anomalies were found in GII and GIII for late dental development and microdontia. However, a type of treatment-dependent division of the study cohort made an analysis difficult. The HR patients (GI) that had undergone intensive treatment and cranial irradiation presented with the lowest number of dental abnormalities. The authors emphasized that there was the lowest number of patients younger than 6 years at the start of the anticancer treatment in this group. It was also discussed that the higher number of teeth abnormalities diagnosed in both remaining groups (GII and GIII) might have been associated with the higher frequency of younger patients, even though radiation was not implemented. Despite the different treatment protocols used by the study participants, the authors finally suggested that the altered dental development is related to the stage of tooth development during anticancer therapy. This is consistent with more common hypoplasia and root disturbances observed in GI and a higher incidence of microdontia and delayed development found in younger groups GII and GIII [21].

In order to establish a similarity of treatment regimens, the duration of particular protocols has been compared. Low standard deviations were noted when it comes to the duration of all analyzed treatment protocols (Table 1). The authors decided to consider two chosen duration parameters: the duration of intensive protocols without breaks for the patient’s back to normal hematological and clinical status affected by chemotherapy and the entire treatment period to exclude factors that could have influenced the study’s result. And as a result of the relationship evaluation, no impact of the duration of the therapy on the number of tooth abnormalities in the study group was noticed. The same treatment duration and different number of dental adverse effects changing over the patient’s age is observed (Table 3). There is an agreement with the opinion presented in some papers that the number of tooth abnormalities does not differ depending on the duration of the therapy [5,11,47]. Unlike treatment duration, the cumulative dose received predominantly varied with the patient’s age or, more precisely, the dose level increased with the patient’s body surface (Table 2). Even so, chemotherapy affected younger patients more severely. No agenesis and reduction in tooth size was diagnosed in older patients except for third molars reduced in size in the oldest patient with treatment onset at the age of 7 years and 7 months. This patient also experienced root abnormalities in canines and premolars. This is consistent with the expected period of developmental stage of these teeth. For the same reason, no root abnormalities were observed in the remaining ALL survivors aged 29–48 months at cancer diagnosis. Patients aged 29–33 months at the treatment onset experienced agenesis of third molars, tooth reduction in the size of the second premolars and second molars, and taurodontic first molars. No abnormal first premolars were found in the study group because their development is expected to be more advanced in patients older than 29 months. Two patients aged 47 and 48 months at the start of the treatment had no typical chemotherapy sequelae. In their previous study, the authors indicated that the age at cancer diagnosis between 36 and 84 is not critical for dental development to be radiographically noticed [48]. This has been confirmed in numerous previous papers, as already mentioned [5,6,10,15,16,17,18,19,20,21,22]. In conclusion, it is not surprising that no significant relationships have been found between the number of abnormalities and the cumulative dose of any of the analyzed drugs in the current study (*p* > 0.050 for all pairs of variables) (Table 4). In the author’s previous study, the number of abnormal teeth was analyzed in relation to single, cumulative, and weekly doses of six drugs administered the most frequently. The study was performed on survivors with different cancer diagnoses. The largest number of abnormalities was observed in patients treated with the highest doses of VCR, DXR, CP, and ACTD when the single doses were analyzed. No similar result was seen in terms of cumulative and weekly doses because the largest number of disturbed teeth was observed after administration of the highest doses of VCR and CP, respectively. Moreover, the majority of patients receiving the least drug doses had a higher or comparable number of dental abnormalities compared to the mean number in each drug group, no matter whether the single, cumulative, or weekly dose were taken into account [15]. According to the results of the current and previous studies, the authors conclude that a noticeable variation in the number of disturbed teeth is strongly dependent on the patient’s age. This has been widely discussed in the literature [5,6,10,17,18,19,20,21,22]. According to Stolze et al., the main risk factor for dental abnormalities occurrence is the age of less than 3 years at cancer diagnosis, despite 51.3% of all study participants being older than 5 years. The authors reported a relatively low incidence of dental abnormalities (36.1%) compared to those found in the other studies, probably due to the small number of young patients who seem to be more sensitive to the toxic effect of chemotherapy [42].

The primary goal of anticancer therapy is to achieve disease remission. Therefore, adverse effects of treatment are inevitable despite efforts to limit the toxic impact of anticancer drugs. Our research shows that their occurrence does not depend on the duration of therapy or the drug dose used in treatment protocols. The treatment duration for leukemia is the longest among all cancer diseases. However, dental changes were observed in patients subjected to much shorter protocols [15,16]. Aware of the inevitability of dental long-term effects, the authors hope that patients will be made aware of the possibility of their occurrence and will be provided with appropriate dental care.

### Limitations

A limitation associated with the research analyzing treatment duration- and dose-dependency of long-term dental effects of the chemotherapy has been overcome. All survivors enrolled in the study presented with the same cancer diagnosis, and they received the therapy according to the same protocol. However, a small study cohort may significantly impact the study result.

## 5. Conclusions

Immature dental structures are susceptible to the antimitotic ability of the drugs used for pre-B ALL treatment. The duration of the chemotherapy did not differ with the age of the study participants, and thereby, this could not have any impact on the various number of tooth abnormalities. No relationship was also found between the level of toxic drug doses and the number of dental sequelae. The age at the onset of antineoplastic therapy is likely to be the strongest risk factor for toxic injury during tooth development. A small study cohort is a major limitation, and an evaluation of similar relationships at larger research centers is needed. Increasing patients’ awareness of dental adverse effects of chemotherapy is also needed.

## Figures and Tables

**Figure 1 cancers-15-05420-f001:**
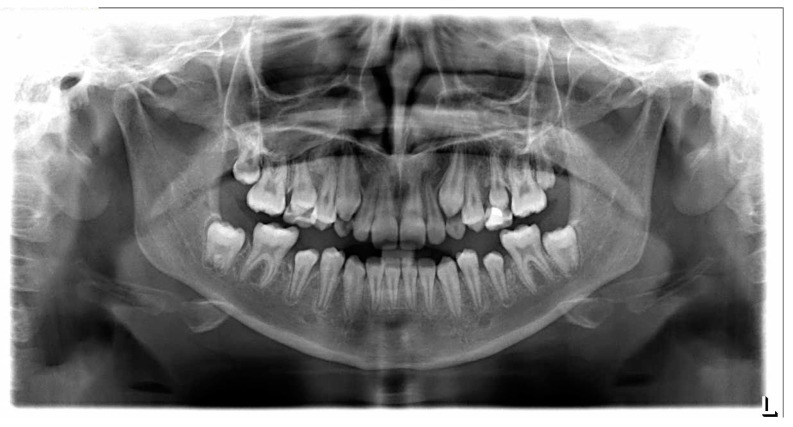
Panoramic radiograph in an 11-year-old female patient. Age at ALL diagnosis: 29 months. All third molars are missing. Teeth 25, 35, 45, 17, 27, 37, and 47 reduced in size can be seen.

**Table 1 cancers-15-05420-t001:** Group characteristics.

Patient’s Age at Diagnosis (Months)	Diagnosis	Treatment Protocol	Age at Dental Examination (Months)		Treatment Duration (Weeks)						Teeth Affected					
				Protocol I	Protocol M	Protocol II	Intensive Therapy (Breaks Excluded)	Maintenance Therapy	Intensive Therapy (Breaks Included)	Entire Therapy (Breaks Included)	Total	Agenesis	Tooth Reduction in Size	Root Abnormalities	Taurodontia	Others
29	ALL SR	ALL IC-BFM 2002, SR	120	10	8	10	28	88	33	122	4		17,27,37,47			
29	ALL IR	ALL IC-BFM 2002, IR	134	11	8	9	28	74	32	112	11	18,28,38,48	25,35,45,17,27,37,47			
29	ALL IR	ALL IC-BFM 2002, IR	92	15	8	6	29	71	33	111	4		15,25,17,27			
33	ALL SR	ALL IC-BFM 2002, SR	112	10	8	5	23	74	31	105	3		27		16,26	
47	ALL SR	ALL IC-BFM 2002, SR	140	12	9	7	28	74	32	104	1					21dens evaginatus
48	ALL SR	ALL IC-BFM 2002, SR	96	21	8	9	28	74	42	118	1					46 tooth impacted
91	ALL SR	ALL IC-BFM 2002, SR	207	11	8	7	26	74	30	111	14		18,28,38,48	14,15,24,25,33,34,35,43,44,45		
M ± SD				12.86 ± 3.98	8.14 ± 0.38	7.57 ± 1.81	27.14 ± 2.04	75.57 ± 0.59	33.29 ± 3.99	111.86 ± 6.47						

ALL—acute lymphoblastic leukemia; SR—standard risk; IR—intermediate risk; M—median; SD—standard deviation.

**Table 2 cancers-15-05420-t002:** Treatment characteristics (ALL IC-BFM 2002).

Patient’s Age at Diagnosis (Months)/Body Surface Along the Therapy (m^2^)	Protocol I							Protocol M			Protocol II							Maintenance Therapy		
Drug M of DA Unit	VCR i.v. mg	DR p.i. mg	LASP p.i. IU	MTX i.th. mg	CP p.i. mg	Ara-C i.v. mg	6-MP p.o. mg	MTX i.th. mg	MTX p.i. g	6-MP p.o. mg	VCR i.v. mg	DXR p.i. mg	LASP p.i. IU	MTX i.th. mg	CP p.i. mg	Ara-C i.v. mg	6-Thio p.o. mg	MTX i.th. mg	MTX p.o. mg	6-MP p.o. mg
	Cumulative doses																			
29 0.6–0.74	3.6	36	4800	40	1200	765	972	48	4.8	700	3.92	78	2600	20	650	390	403.2	48	1100	23,025
29 0.6–0.74	3.6	72	4800	50	1200	720	1008	48	4.8	700	1.8	72	2400	20	600	360	416	48	380	7175
29 0.6–0.65	3.6	72	24,000	20	1200	720	1548	40	4.8	955	3.92	78	26,000	20	650	390	468	40	740	12,775
33 0.5–0.55	3	60	4000	60	500	-	-	48	4	-	3	60	2000	-	500	-	-	-	740	12,950
47 0.75–0.85	4.5	45	6000	60	750	900	-	48	6.4	2400	5.4	96	3200	24	800	480	560	48	7770	19,425
48 0.7–0.75	4.2	42	8750	24	1400	630	1176	40	5.6	850	4.2	84	56,000	-	700	420	-	40	970	18,025
91 1.05–1.1	6.3	126	8400	48	2100	1260	-	48	8.4	-	6.6	132	4400	24	1100	660	440	48	10,360	25,900
	Single doses																			
29	0.9	18	600	10	600	180	36	12	1.2	36	0.98	19.5	650	10	650	48.75	33.6	12	12.5	37.5
29	0.9	18	600	10	600	180	36	10	1.2	12.5	0.45	18	600	10	600	45	26	12	5 *	12.5 *
29	0.9	18	3000	10	600	180	36	10	1.2	12.5	0.98	19.5	6500	10	650	48.75	36	10	10 *	25 *
33	0.75	15	500	12	500	-	-	12	1	-	0.75	15	500	-	500	37.5	-	-	10	25
47	1.13	22.5	750	12	750	225	-	12	1.6	37.5	1.35	24	800	12	800	60	40	12	15	37.5
48	1.05	21	1750	12	700	210	-	10	1.4	12.5	1.05	21	7000	-	700	52.5	-	10	15 $	37.5 $
91	1.58	31.5	1050	12	1050	315	-	12	2.1	-	1.65	33	1100	12	1100	82.5	44	12	20	50
Treatment details	4 doses	2–4 doses	6–8 doses	2–5 doses	1–2 doses	4 doses	28–43 days	4 doses	4 doses	43–68 days	4 doses	4 doses	4–8 doses	6–7 dose	1 dose	2 doses	11–16 days	4 doses	every week	every day
	1-day cycle weekly	1-day cycle weekly	1-day cycle every 3 day	1-day cycle	1-day cycle	4–5-day cycle weekly	1-day cycle every day	1-day cycle every 2 weeks	1-day cycle every 2 weeks	daily	1-day cycle weekly	1-day cycle weekly	1-day cycle every 3 day	1-day cycle every 1–2 week	1-day cycle	4-day cycle weekly	daily	1-day cycle every 4 week	weekly	daily

VCR—vincristine; DR—daunorubicin; LASP—L-asparaginase; MTX—methotrexate; CP—cyclophosphamide; Ara-C—cytarabine; 6-MP—6-Mercaptopurine; DXR—doxorubicin; 6-Thio—6-thioguanine; M of DA (method of drug administration): i.v.—intravenous push; p.i.—intravenous infusion; i.th.—intrathecal; p.o.—by mouth. * The dose was decreased after a few weeks of treatment. $ The dose was increased after a few weeks of treatment.

**Table 3 cancers-15-05420-t003:** Correlation between treatment duration of intensive therapy (breaks excluded), treatment duration of the entire therapy (breaks included), and the number of tooth abnormalities.

Patient’s Age at Diagnosis Months	Treatment Duration of Intensive Therapy (Breaks Excluded) Weeks	Number of Tooth Abnormalities	Treatment Duration of the Entire Therapy (Breaks Included) Weeks	Number of Tooth Abnormalities
29	28	4	122	4
29	28	11	112	11
29	29	4	111	4
33	23	3	105	3
47	28	1	104	1
48	28	1	118	1
91	26	14	111	14
tau-b	−0.12		0.21	
*p*-value	0.732		0.534	

tau-b—Kendall’s correlation coefficient.

**Table 4 cancers-15-05420-t004:** The relationship between the cumulative dose of the analyzed drugs and the number of tooth abnormalities within the study group.

Patient’s Age at Diagnosis (Months)	VCR i.v. mg	N	LASP i.v. IU	N	MTX i.th. mg	N	CP p.i. mg	N	Ara-C i.v. mg	N	6-MP p.o. mg	N
29	7.52	4	7400	4	156	4	1850	4	1155	4	23,725	4
29	5.4	11	7200	11	166	11	1800	11	1080	11	7875	11
29	7.52	4	50,000	4	120	4	1850	4	1110	4	13,730	4
33	6	3	6000	3	108	3	1000	3	-	3	12,950	3
47	9.9	1	9200	1	180	1	1550	1	1380	1	21,825	1
48	8.4	1	64,750	1	104	1	2100	1	1050	1	18,875	1
91	12.9	14	12,800	14	168	14	3200	14	1920	14	25,900	14
tau-b	−0.15		−0.15		0.45		0.26		0.22		0.05	
*p*-value	0.641		0.645		0.167		0.437		0.559		0.878	

N—number of tooth abnormalities; tau-b—Kendall’s correlation coefficient.

## Data Availability

All data are included in the study in the form of mean values. Detailed information is available upon request from the authors.

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
