# Peer review of "Tooth Abnormalities and Their Age-Dependent Occurrence in Leukemia Survivors"

_cancers, 2023, doi:10.3390/cancers15225420_

Round 1

Reviewer 1 Report

Comments and Suggestions for Authors

Dear Authors,

 The manuscript entitled "Tooth abnormalities and their age-dependent occurrence in leukemia survivors" is very interesting. The manuscript presented scientifically sound arguments on the importance of effective research.

1.     Studies that were previously published are cited, but most of the citations are older than 5 years.

2.     The phrase "all cancer survivors after caregivers have given written consent to participate in the study" (lines 164–165) is generally located at the beginning of the materials and methods subsection, after the study approval number (Bioethics Committee at the Medical University of Silesia in Katowice… ).

3.     The sentence "The purpose of the study was to establish a relationship between the duration of treatment and the number of abnormalities" (lines 171–172) is generally located at the end of the introduction and contains the same idea as in the abstract subsection.

4.     In the results subsection, it is necessary to present a table showing only the location and types of developmental anomalies observed in the teeth of the selected surviving patients.

5.     The discussion subsection is too long.

6.     The clinical relevance of the study should be mentioned before the conclusions subsection.

7.     The cited references are relevant, but of a total of 48 references, only 13 (references 1, 4, 15, 16, 18, 28, 29, 32, 34, 35, 37, 42, 48) were published in the past 5 years.

8.     Be careful with text editing (e.g., punctuation in text, line 137).

 For the reasons given above, I consider that the manuscript deserves to be published, but after making major corrections.

Author Response

Anna Jodłowska, PhD

Department of Pediatric Dentistry, Medical University of Silesia in Katowice, Poland, 41-800 Zabrze, Pl. Traugutta 2

Lidia Postek-Stefańska, PhD with habilitation

Department of Pediatric Dentistry, Medical University of Silesia in Katowice, Poland, 41-800 Zabrze, Pl. Traugutta 2

November 2, 2023

Refers to: Manuscript ID: cancers-2628352
Type of manuscript: Article
Title: "Tooth abnormalities and their age dependent occurrence in leukemia survivors"
Special Issue: "Side Effects of Anticancer Therapy: Prevention and Management"

Dear Reviewer,

First of all, we would like to thank the Reviewer for invaluable contribution to the creation of the manuscript. We are very pleased that Reviewer found the study interesting and very necessary.  All the Reviewer’s comments demonstrated again a thorough reading and analysis of our work, for which we are very grateful. The authors took a second look at the manuscript and made changes where possible without compromising the value of the study. The authors kindly ask for understanding in maintaining their own opinion regarding certain comments from the reviewers and ask for considering publishing this paper in the journal “Cancers”. We will repeat that it is the unique opportunity to consider the toxicity of the drugs from an oncological point of view, as well as the complicated mechanism of tooth development from a dental point of view.

As requested, I am including a reference to the comments below.

 The manuscript entitled "Tooth abnormalities and their age-dependent occurrence in leukemia survivors" is very interesting. The manuscript presented scientifically sound arguments on the importance of effective research.

  1. Studies that were previously published are cited, but most of the citations are older than 5 years.

Answer 1:

The authors are perfectly aware of the importance of the latest studies and therefore literature for the development of science and progress in medicine. Starting from the research on the side effects of anticancer therapy to the relationships between their occurrence and individual risk factors, the topic discussed in the manuscript is not very widespread in the literature. Whether, in the connection with the above, we should withdraw from dealing with this topic? The authors noticed a huge gap in the literature on this topic, as well as incorrect assumptions made in some of the few recent works. This is why the authors so often cite their own papers, which does not make the conclusions discussed very objective. We would like to encourage similar studies to be undertaken in larger centers to confirm the thesis about toxic effect of therapy depending on the patient’a age, provided that the impact of too many factors on the study results is limited.

  1. The phrase "all cancer survivors after caregivers have given written consent to participate in the study" (lines 164–165) is generally located at the beginning of the materials and methods subsection, after the study approval number (Bioethics Committee at the Medical University of Silesia in Katowice… ).

Answer 2:

Thank you for this comment, because thanks to it we noticed the missing information about the dental examination performed resulting from the previous revision. The missing information has already been included in the text of the paper, as can be seen below.

Previous text: “ALL cancer survivors after caregivers gave their written consent for the participation in the study. Thorough analysis revealed a total of 37 teeth with different developmental anomalies and one tooth failed to erupt (Figure 1). Detailed information is summarized in Table 1. “

Revised text: “ALL cancer survivors, after caregivers gave their written consent for the participation in the study, have undergone a dental examination, including panoramic radiograph (line 165). Thorough analysis revealed a total of 37 teeth with different developmental anomalies and one tooth failed to erupt (Figure 1). Detailed information is summarized in Table 1. “

When it comes to the Reviewer’s indication, to avoid repetition (see citation 15), information about the approval of Bioethics Committee was removed from the paper in the course of the previous revisions. Due to the concept of focusing the reader’s attention on the characteristics of the disease and its treatment, it is very difficult to place information about the approval of the Bioethics Committee at the beginning of the chapter. For this reason the authors resorted only to citing the relevant literature (citation 15).  Moreover, the subject of the current study is an analysis based on the results presented in the previous paper (citation 15) and data from medical records of survivors. The topic of the research is not directly dependent on the committee decision. If the information about the approval of Bioethics Committee were to return to the Material and methods, the authors propose this wording of the text, which can be seen in the simulation shown below.

Previous text: “A group of 40 cancer survivors attended to Outpatient Dental Clinic following the inclusion criteria established in the previous cross-sectional study [15]: age of 5-18 years and minimum 2 years after anticancer therapy completion. The patients were in remission and they were monitored regularly in their Outpatient Hematology and Oncology Clinic. An attempt to identify pediatric leukemia survivors was made. Of 8 leukemia survivors 7 participants diagnosed with pre-B ALL code 204.0 according to International Classification of Diseases (ICD-9) have been selected. They were treated under similar conditions in the same hospital between 2005 and 2013. The anticancer therapy was carried out according to similar protocols depending on the risk of malignancy. Three of 7 individuals were classified as having SR-ALL and 4 patients presented with HR disease. ALL IC-BFM 2002 protocol was administered to all of them with small variations depending on the risk diagnosed. The medical records of all survivors were thoroughly reviewed in terms of drugs administered, their doses and treatment schedules. All data have been presented in Table 1 and Table 2. At the moment of enrolling pre-B ALL patients we observed 24-month event-free survival minimum and 78-month event-free survival maximum after the maintenance therapy completion in 6 selected individuals. One participant developed two bone marrow relapses after he had 18-month and 14.5-month event-free periods, respectively. The new different protocols were introduced to him (REZ BFM 2002 and INT ReAll 2010, respectively), but the treatment did not overlapped with the critical developmental period of his dentition. That is why the patient was not excluded from the study and both of additional regimens have not been undertaken into comparative analysis. Relapsed survivor received prophylactic central nervous system radiotherapy (CSN RT) when aged 12 and stem cell transplantation (SCT) following the last course of chemotherapy at the age of 17. Neither RT nor SCT was given into the remaining study participants.”

Simulation: “A group of 40 cancer survivors attended to Outpatient Dental Clinic following the inclusion criteria established in the previous cross-sectional study [15]: age of 5-18 years and minimum 2 years after anticancer therapy completion. The previous study was approved by the Bioethics Committee of the Medical University of Silesia on February 25, 2013 and on November 29, 2016 (KNW/0022/KB1/15/I/13,  KNW/0022/KB1/15/II/16). The patients were in remission and they were monitored regularly in their Outpatient Hematology and Oncology Clinic. An attempt to identify pediatric leukemia survivors was made. Of 8 leukemia survivors 7 participants diagnosed with pre-B ALL code 204.0 according to International Classification of Diseases (ICD-9) have been selected. They were treated under similar conditions in the same hospital between 2005 and 2013. The anticancer therapy was carried out according to similar protocols depending on the risk of malignancy. Three of 7 individuals were classified as having SR-ALL and 4 patients presented with HR disease. ALL IC-BFM 2002 protocol was administered to all of them with small variations depending on the risk diagnosed. The medical records of all survivors were thoroughly reviewed in terms of drugs administered, their doses and treatment schedules. All data have been presented in Table 1 and Table 2. At the moment of enrolling pre-B ALL patients we observed 24-month event-free survival minimum and 78-month event-free survival maximum after the maintenance therapy completion in 6 selected individuals. One participant developed two bone marrow relapses after he had 18-month and 14.5-month event-free periods, respectively. The new different protocols were introduced to him (REZ BFM 2002 and INT ReAll 2010, respectively), but the treatment did not overlapped with the critical developmental period of his dentition. That is why the patient was not excluded from the study and both of additional regimens have not been undertaken into comparative analysis. Relapsed survivor received prophylactic central nervous system radiotherapy (CSN RT) when aged 12 and stem cell transplantation (SCT) following the last course of chemotherapy at the age of 17. Neither RT nor SCT was given into the remaining study participants.

  1. The sentence "The purpose of the study was to establish a relationship between the duration of treatment and the number of abnormalities" (lines 171–172) is generally located at the end of the introduction and contains the same idea as in the abstract subsection.

Answer 3:

Of course, another wording of this sentence is more applicable. The new sentence has been placed in the text.

Previous text: “The aim of the study was to establish a relationship between treatment duration and number of abnormalities. A special attention was paid to the duration of a real period during which the anticancer drugs were taken by patients. For this purpose, the duration of intensive therapy after excluding all the treatment breaks was determined. Additionally, the duration of the entire therapy was also taken into correlation analysis (Table 3).”

Revised text: “In order to establish a relationship between treatment duration and number of dental abnormalities,  the duration of a real period during which the anticancer drugs were taken by patients was calculated( lines 171-173). For this purpose, the duration of intensive therapy after excluding all the treatment breaks was determined. Additionally, the duration of the entire therapy was also taken into correlation analysis (Table 3).”

  1. In the results subsection, it is necessary to present a table showing only the location and types of developmental anomalies observed in the teeth of the selected surviving patients.

Answer 4:

Unfortunately, the note is not understandable to the authors. Perhaps the misunderstanding stems from the fact that the authors had problems with correct editing of the table. In the Material and methods the tables containing data needed to achieve the goal of the study were included. Among others, it contains Table 1 showing the location and types of dental anomalies in selected survivors. The details included in Table 1 allow only for the justification of the toxic effect of treatment on tooth germs. In fact, only the number of abnormalities was used as a variable in the relationship establishment. Moreover, the authors made sure that the Results contained data strictly related to the purpose of the study.

  1. The discussion subsection is too long.

Answer 5:

The authors realize that the chapter is too long. In our opinion the text in the lines 222-336 is difficult to read, although every effort has been made to ensure the clearest arrangement of the issues discussed. The authors decided to remove only a fragment 256-258 as it was irrelevant to the study. Due to many doubts regarding the cited literature, the authors tried to provide justification for the need to undertake research on this topic. Due to limited literature from the past 5 years cited in the study, it is even more difficult to limit the content in this part of the Discussion. Could you please take a look at the excerpts below.

“Although the dental developmental abnormalities as the anticancer treatment effects have been widely documented, the type of the therapy was usually not analyzed in the past” (lines 225-227).

“The influence of the drugs and their doses has not been taken into study consideration” (line 236)

“In the paper there is only an information on drug dose seemed to be a single one because the dosing method such as mg/m2 has been supplemented.” (lines 249-250)

“However, the study was based on self-reported questionnaire what seems to be a strong limitation” (lines 263-264)

“However, the authors indicate a major limitation of the study, as the historical data were obtained from the dentists without the possibility of X-ray analysis” (lines 266-268)

”Although ALL and Wilms tumor were the most frequently diagnosed conditions in the study participants, the impact of many concomitant factors cannot be excluded” (lines 271-273)

“So common in the literature the method of evaluating a toxicity of single therapeutic agent given in the multidrug protocol was considered the strong limitation” (lines 290-292)

“Wilberg et al. also met some limitations in their study and saw the need to assess a large sample size with a single diagnosis, relatively homogenous therapy and long observation time.” (lines 294-296)

“Some authors have examined pediatric cohorts with the same cancer diagnosis, but they reported the prevalence of dental abnormalities having no interest in terms of treatment influencing factors” (lines 296-298)

“No treatment details were taken into consideration.” (lines 303-304)

“There were no dental abnormalities observed in the study” (lines 306-307)

“There were no investigation on the relationship between treatment conditions and dental abnormalities occurrence in the study” (lines 309-310)

“…but there are no statistical results relating to the treatment duration and individual drug doses in the paper.“(lines 316-317).

A widely discussed in the paper an interesting research of Minicucci et al. (lines 317-336) is very useful for the authors due to the justification of the dependence of the occurrence of particular developmental anomalies on the patient’s age at cancer diagnosis. Thanks to similar research, we can link diagnosed developmental abnormalities with anticancer therapy, which may be questionable in a small study cohort.

The remaining part of the chapter is an analysis of the assumptions made by the authors as the aim of the study. Deleting any sentence would be detrimental to the study.

  1. The clinical relevance of the study should be mentioned before the conclusions subsection.

Answer 6:

This is a very valuable comment due to the low awareness of dental side effects in the oncology community. Our patients were not informed about the possibility of their occurrence after anticancer therapy. In our opinion this publication adequately covers this aspect of the issue. However, extending this issue to include the influence of drug dose or patient’s age on the occurrence of possible dental abnormalities goes beyond the topic of the presented study. Possible speculations require referring to the research results presented in our previous publications, which were extensively discussed here anyway due to the lack of appropriate literature. Moreover, any mention of the occurrence of other long-term effects and the drugs used to reduce their severity was removed from manuscript during the previous edition of the text (reviewer’s recommendation).  This information would now be useful as an introduction to the desired topic. For that reason, we will basically limit the information to raising the awareness of oncologists about dental toxic effect of anticancer therapy. A new paragraph has been placed at the end of Discussion.

“The primary goal of anticancer therapy is to achieve disease remission. Therefore, adverse effects of treatment are inevitable, despite efforts to limit toxic impact of anticancer drugs. Our research shows that their occurrence does not depend on the duration of therapy or drug dose used in treatment protocols. The treatment duration for leukemia is the longest among all cancer diseases. However, dental changes were observed in patients subjected to much shorter protocols [15,16]. Aware of the inevitability of dental long-term effects, the authors hope that patients will be made aware of the possibility of their occurrence and will be provided with appropriate dental care.”

  1. The cited references are relevant, but of a total of 48 references, only 13 (references 1, 4, 15, 16, 18, 28, 29, 32, 34, 35, 37, 42, 48) were published in the past 5 years.

Answer 7:

        We have already referred to the above issue in Answer 1. We can only add that when undertaking this study, we were aware that it would be very difficult to discuss this topic. First of all, we wanted to pay special attention to the trend of conducting research on very diverse cohorts noticed in the literature. At the same time, the authors want to emphasize that this is definitely due to the difficulty in assembling appropriate study groups, as evidenced by the limited literature on this topic and limitations in the cited studies presented in the Answer 5.

  1. Be careful with text editing (e.g., punctuation in text, line 137).

Answer 8:

We are very grateful for this important advise. Changes were introduced by the authors as follows.

Previous sentence: “The objective of the study was to establish a relationship between the duration and the dose of chemotherapy, and number of tooth abnormalities in pre-B All survivors receiving the treatment according to the same protocol.”

Revised sentence:The objective of the study was to establish a relationship between the duration of treatment, the dose of chemotherapy and number of tooth abnormalities in pre-B All survivors receiving the treatment according to the same protocol.”

Your Sincerely,

Dr. Anna Jodłowska

Reviewer 2 Report

Comments and Suggestions for Authors

Although the authors have made substantial revisions to the manuscript, improving the quality of the article, a central question still remains highly questionable: how to establish with precision a relationship between the treatment duration and the number of dental abnormalities, inferring that this is a result of the treatment rather than a potential congenital nature?

Author Response

Anna Jodłowska, PhD

Department of Pediatric Dentistry, Medical University of Silesia in Katowice, Poland, 41-800 Zabrze, Pl. Traugutta 2

Lidia Postek-Stefańska, PhD with habilitation

Department of Pediatric Dentistry, Medical University of Silesia in Katowice, Poland, 41-800 Zabrze, Pl. Traugutta 2

November 2, 2023

Refers to: Manuscript ID: cancers-2628352
Type of manuscript: Article
Title: "Tooth abnormalities and their age dependent occurrence in leukemia survivors"
Special Issue: "Side Effects of Anticancer Therapy: Prevention and Management"

Dear Reviewer,

First of all, we would like to thank the Reviewer for invaluable contribution to the creation of the manuscript. We are very pleased that Reviewer found the study interesting and very necessary.  All the Reviewer’s comments demonstrated again a thorough reading and analysis of our work, for which we are very grateful. The authors took a second look at the manuscript and made changes where possible without compromising the value of the study. The authors kindly ask for understanding in maintaining their own opinion regarding certain comments from the reviewers and ask for considering publishing this paper in the journal “Cancers”. We will repeat that it is the unique opportunity to consider the toxicity of the drugs from an oncological point of view, as well as the complicated mechanism of tooth development from a dental point of view.

As requested, I am including a reference to the comments below.

Although the authors have made substantial revisions to the manuscript, improving the quality of the article, a central question still remains highly questionable: how to establish with precision a relationship between the treatment duration and the number of dental abnormalities, inferring that this is a result of the treatment rather than a potential congenital nature?

Answer :

We are aware that a study based on such a small number of participants does not justify such bold conclusions. Therefore, in our paper we widely present the results of our previous studies [15,16] on a larger group of cancer survivors, from which leukemia survivors were selected. Moreover, extensive literature on dental long-term effects of anticancer therapy is cited in the current paper. The literature is old, which is the reason for the valid comments of the reviewers, but it proves that the topic has been well documented in the literature. To establish the aimed rtelationships, the authors used only the number of dental abnormalities. However, Table 1 provides detailed data on the types of dental anomalies and the teeth abnormal. Based on this data the following conclusion can be made: the developmental period of individual teeth presenting particular anomalies coincides with the age at anticancer treatment. The authors mention this in Introduction and in Discussion (information below).

The time of the occurrence of an individual dental abnormality is precisely determined by historical literature. The coincidence of the age at cancer diagnosis with the stage of tooth development, the toxic damage of which leads to the anomaly, is not accidental. Moreover, no such significant developmental abnormalities in the crown of second premolars and second molars are found in healthy population. For example, no agenesis and microdontia of first premolars are diagnosed in healthy individuals, but in cancers survivors, but younger than presented in the study, a great number of abnormal first premolars are noted. Due to the huge amount of the literature on this topic, the authors did not take this problem into consideration in the current study.

“It has been well documented that exposure to chemotherapy always causes side effects in the human body [1-3]. Their occurrence depends on the number of factors, sometimes unknown. There is strong evidence that dental abnormalities are late adverse effects of antineoplastic therapy when it is introduced at a young age [4-14].”

“It has been already proven that young cancer survivors presenting with the most severe dental changes had been receiving chemotherapy before they were 3-5 years old. Furthermore, they exhibited no dental abnormalities if they had been treated at the age between 43 and 61 months [5,6,10,15-22].”

“Although the dental developmental abnormalities as the anticancer treatment effects have been widely documented, the type of the therapy was usually not analyzed in the past [4,7,19,20,22,40,41]. Many risk factors have been considered but the age at the onset of the therapy was the most frequently quoted [4-6,10,17-22,42]. Cubukcu et al. emphasize that the same treatment protocol may cause different changes in different teeth depending on the stage of development [7]. Although the dental developmental stage thereby patient’s age at the start of the anticancer therapy seems to be the strongest risk factor, the impact of treatment duration and particular toxic agents has been analyzed in some recent papers [5,15,16,23,42-44]. Proc et al. confirmed that the chemotherapy has a significant impact on odontogenesis, but an established number of tooth abnormalities did not differ with treatment duration.”

“Rosenberg et al. studied 17 long-term survivors of childhood ALL for altered dental root development and no treatment analysis was conducted [14]. In another study 52 ALL and acute myeloid leukemia (AML) survivors treated according to the same protocol referred as AIEOP (Italian Association of Paediatric Hematoncology) were controlled in order to dental sequelae assessment. No treatment details were taken into consideration [24].”

“…Despite the different treatment protocols used in the study participants, the authors finally suggested that the altered dental development is related to the stage of tooth development during anticancer therapy. This is consistent with more common hypoplasia and root disturbances observed in GI, and a higher incidence of microdontia and delayed development found in younger groups GII and GIII [21].”

“And as the result of relationship evaluation no impact of the duration of the therapy on the number of tooth abnormalities in the study group was noticed. The same treatment duration and different number of dental adverse effects changing over the patient’s age is observed (Table 3). There is in agreement with the opinion presented in some papers that number of tooth abnormalities do not differ depending on the duration of the therapy [5,11,47]. Unlike treatment duration, the cumulative dose received predominantly varied with the patient’s age or, more precisely, the dose level increased with the patient’s body surface (Table 2). Even so, chemotherapy affected younger patients more severely. No agenesis and reduction in tooth size were diagnosed in older patients except for third molars reduced in size in the oldest patient with treatment onset at the age of 7 years and 7 months. This patient also experienced root abnormalities in canines and premolars. This is consistent with the expected period of developmental stage of these teeth. For the same reason no root abnormalities were observed in the remaining ALL survivors aged 29-48 months at cancer diagnosis. Patients aged 29-33 months at the treatment onset experienced agenesis of third molars, tooth reduction in size of second premolars and second molars, and taurodontic first molars. No abnormal first premolars were found in the study group because their development is expected to be more advanced in patients older than 29 months. Two patients aged 47 and 48 months at the start of the treatment had no typical chemotherapy sequelae. In their previous study the authors indicated the age at cancer diagnosis between 36 and 84 as not critical for dental development to be radiographically noticed [48]. This has been confirmed in numerous previous papers, as already mentioned [5,6,10,15-22].”

Your Sincerely,

Dr. Anna Jodłowska

Reviewer 3 Report

Comments and Suggestions for Authors

The authors investigated the impact of chemotherapy on tooth abnormalities in a small cohort of children previously treated for ALL. They suggest that the first cause of dental abnormalities is age of treatment.

The paper is interesting and underlines the importance of considering late side effect of chemo-regimens. 

I Have just a question: 

Why did they exclude that the small size of studied population is the cause of the lack of statistical difference ? Di they consider the possible cumulative impact of the drug known to negative impact developing tissues? 

Minor point: please check risk definition in the text, page 3 line 149, and in table 1. Data do not match.

Author Response

Anna Jodłowska, PhD

Department of Pediatric Dentistry, Medical University of Silesia in Katowice, Poland, 41-800 Zabrze, Pl. Traugutta 2

Lidia Postek-Stefańska, PhD with habilitation

Department of Pediatric Dentistry, Medical University of Silesia in Katowice, Poland, 41-800 Zabrze, Pl. Traugutta 2

November 2, 2023

Refers to: Manuscript ID: cancers-2628352
Type of manuscript: Article
Title: "Tooth abnormalities and their age dependent occurrence in leukemia survivors"
Special Issue: "Side Effects of Anticancer Therapy: Prevention and Management"

Dear Reviewer,

First of all, we would like to thank the Reviewer for invaluable contribution to the creation of the manuscript. We are very pleased that Reviewer found the study interesting and very necessary.  All the Reviewer’s comments demonstrated again a thorough reading and analysis of our work, for which we are very grateful. The authors took a second look at the manuscript and made changes where possible without compromising the value of the study. The authors kindly ask for understanding in maintaining their own opinion regarding certain comments from the reviewers and ask for considering publishing this paper in the journal “Cancers”. We will repeat that it is the unique opportunity to consider the toxicity of the drugs from an oncological point of view, as well as the complicated mechanism of tooth development from a dental point of view.

As requested, I am including a reference to the comments below.

The authors investigated the impact of chemotherapy on tooth abnormalities in a small cohort of children previously treated for ALL. They suggest that the first cause of dental abnormalities is age of treatment.

The paper is interesting and underlines the importance of considering late side effect of chemo-regimens. 

I Have just a question: 

Why did they exclude that the small size of studied population is the cause of the lack of statistical difference ? Di they consider the possible cumulative impact of the drug known to negative impact developing tissues? 

Minor point: please check risk definition in the text, page 3 line 149, and in table 1. Data do not match.

Answer:

  1. The first question is very difficult for us to understand. There is no information regarding statistical differences in any of the analyzed variables in the paper. A very valuable comment would be why the authors assumed that such a small group could be representative of reliable research. We assume this may have been what the reviewer had in mind when he asked this question. If yes, this question is very valid. The authors were afraid of such an accusation, so they determined a standard deviation for the study group in terms of treatment duration of each treatment protocol, although it is common knowledge that anticancer treatment is carried out according to a strictly defined regimen. The appearance of acute adverse effects sometimes leads to changes in drug administration, but this did not apply to our study cohort. Perhaps the information about statistical analysis turned out to be misleading. For obvious reasons it was not executed. The protocols turned out to be very similar when it comes to their duration. Therefore, the authors decided to remove this information from the manuscript as follows:

Recent text: “In order to establish a similarity of treatment regimens the duration of particular protocols has been statistically compared. A low standard deviations were noted when it comes to the duration of all analyzed treatment protocols (Table 1).”

Revised text: “In order to establish a similarity of treatment regimens the duration of particular protocols has been statistically compared. A low standard deviations were noted when it comes to the duration of all analyzed treatment protocols (Table 1).”

Moreover, the authors would like to draw attention to two fragments of the publication:  Jodłowska, A.; Postek-Stefańska, L. Duration and dose of chemotherapy and dental development. (Dent. Med. Probl. 2022,59,45–58), in which a total of 37 different cancer survivors were subjected to dental examination and statistical analysis was applied to treatment duration and drug dose data.

“No significant differences in the treatment duration were noted between abnormalities within the affected and non-affected groups for each analyzed medication, except for patients with a reduction in tooth size and enamel changes in the CP affected group (p = 0.05) (Table 2). Furthermore, there were no significant differences in the treatment duration between the affected and non-affected patients within most groups of dental abnormalities for each drug, except for microdontia in the DXR group (p = 0.04) and a reduction in tooth size in the CP group (p = 0.03) (Table 2). However, CP therapy in the participants with a reduced crown size was paradoxically longer in the non-affected group, with a similar anomaly appearing in many groups of abnormalities regardless of the drug administered (Table 2).”

“To test the hypothesis that the observed shorter treatment intervals in the first 10 weeks would result in a higher cumulative treatment dose than in the remaining longer period of therapy, a separate calculation was performed for the initial 10 weeks and for the total duration of therapy. There were no statistically significant differences between patients with different disturbances in most specific drug groups when analyzing the treatment duration and the drug doses, except for VCR patients with agenesis and microdontia in the first 10 weeks of therapy.”

Moreover, we would like to inform that the aim of the study was to establish statistically significant relationships mentioned below:

Results:

”No statistically significant correlation was observed between treatment duration of an intensive therapy (breaks excluded), treatment duration of the entire therapy (breaks included) and the number of tooth abnormalities (tau-b = -0.12; p = 0.732 for intensive therapy; tau-b = 0.21; p = 0.534 for the entire therapy) (Table 3).”

“No statistically significant relationship was found between the number of dental abnormalities and the cumulative dose of VCR, LASP, MTX, CP, Cyta, 6-MP (p > 0.050 for all pairs of variables) (Table 4).”

Discussion:

“In order to establish a similarity of treatment regimens the duration of particular protocols has been statistically compared. A low standard deviations were noted when it comes to the duration of all analyzed treatment protocols (Table 1). The authors decided to take into consideration two chosen duration parameters: the duration of intensive protocols without breaks for patient’s back to normal hematological and clinical status affected by chemotherapy and the entire treatment period to exclude factors that could have had some influence on the result of the study. And as the result of relationship evaluation no impact of the duration of the therapy on the number of tooth abnormalities in the study group was noticed. The same treatment duration and different number of dental adverse effects changing over the patient’s age is observed (Table 3). There is in agreement with the opinion presented in some papers that number of tooth abnormalities do not differ depending on the duration of the therapy [5,11,47]. Unlike treatment duration, the cumulative dose received predominantly varied with the patient’s age or, more precisely, the dose level increased with the patient’s body surface (Table 2). Even so, chemotherapy affected younger patients more severely. No agenesis and reduction in tooth size were diagnosed in older patients except for third molars reduced in size in the oldest patient with treatment onset at the age of 7 years and 7 months. This patient also experienced root abnormalities in canines and premolars. This is consistent with the expected period of developmental stage of these teeth. For the same reason no root abnormalities were observed in the remaining ALL survivors aged 29-48 months at cancer diagnosis. Patients aged 29-33 months at the treatment onset experienced agenesis of third molars, tooth reduction in size of second premolars and second molars, and taurodontic first molars. No abnormal first premolars were found in the study group because their development is expected to be more advanced in patients older than 29 months. Two patients aged 47 and 48 months at the start of the treatment had no typical chemotherapy sequelae. In their previous study the authors indicated the age at cancer diagnosis between 36 and 84 as not critical for dental development to be radiographically noticed [48]. This has been confirmed in numerous previous papers, as already mentioned [5,6,10,15-22]. In conclusion, it is not surprising that no significant relationships have been found between the number of abnormalities and the cumulative dose of any of the analyzed drugs in the current study (p > 0.050 for all pairs of variables) (Table 4).”

  1. When it comes to a second question, the authors believe that there was an unintentional error during sending a revision text because it is difficult to understand the reviewer’s suggestions. Negative impact of all anticancer toxic drugs is well documented on animal model and, as a rule, cytostatics stop cell division or cause cell apoptosis. All the more so when administered in the multidrug protocol, they definitely contribute to impared development of tooth germs if used at a young age. On the other hand, if we assume that any of the drugs administerd does not have a strongly toxic effect on developing tissues or the impact is transient, it is still dificult to assess when multidrug protocols are usually realised. The authors attempted to assess the toxic effect of single drugs administerd in multidrug protocols in previous papers and obtained expected results, which prove that a similar analysis does not make sense (kindly see citation 15,16).

  1. We are very grateful for the last comment. We did not noticed this obvious inaccuracy. We would like to apologize for such carelessness. We obviously have made appropriate corrections both in the text and in Table 1.

Previous text: “Three of 7 individuals were classified as having SR-ALL and 4 patients presented with HR disease. ALL IC-BFM 2002 protocol was administered to all of them with small variations depending on the risk diagnosed.”

Revised text: “Five of 7 individuals were classified as having SR-ALL and 2 patients presented with IR disease. ALL IC-BFM 2002 protocol was administered to all of them with small variations depending on the risk diagnosed.”

Your Sincerely,

Dr. Anna Jodłowska

Round 2

Reviewer 1 Report

Comments and Suggestions for Authors

Dear Authors,

 The manuscript entitled "Tooth abnormalities and their age-dependent occurrence in leukemia survivors" is very interesting. 1. Studies that were previously published are cited, but most of the citations are older than 5 years. The cited references are relevant, but of a total of 48 references, only 13 (references 1, 4, 15, 16, 18, 28, 29, 32, 34, 35, 37, 42, 48) were published in the past 5 years.

For the reasons given above, I consider that the manuscript deserves to be published, but after making minor corrections.

Author Response

Anna Jodłowska, PhD

Department of Pediatric Dentistry, Medical University of Silesia in Katowice, Poland, 41-800 Zabrze, Pl. Traugutta 2

Lidia Postek-Stefańska, PhD with habilitation

Department of Pediatric Dentistry, Medical University of Silesia in Katowice, Poland, 41-800 Zabrze, Pl. Traugutta 2

November 12, 2023

Refers to: Manuscript ID: cancers-2628352
Type of manuscript: Article
Title: "Tooth abnormalities and their age dependent occurrence in leukemia survivors"
Special Issue: "Side Effects of Anticancer Therapy: Prevention and Management"

Dear Reviewer,

The authors kindly thank the Reviewer for acceptance of our last version of manuscript and understanding the reasons why certain conditions cannot be met. The authors took a look at some references regarding the topic covered in the current manuscript. The result of the search is contained in the response to the request.

 The manuscript entitled "Tooth abnormalities and their age-dependent occurrence in leukemia survivors" is very interesting. 1. Studies that were previously published are cited, but most of the citations are older than 5 years. The cited references are relevant, but of a total of 48 references, only 13 (references 1, 4, 15, 16, 18, 28, 29, 32, 34, 35, 37, 42, 48) were published in the past 5 years.

For the reasons given above, I consider that the manuscript deserves to be published, but after making minor corrections.

Response:

During our research, we collected extensive literature on late dental adverse effects of antineoplastic treatment. Unfortunately, most of the literature is older than 5 years. Relatively newest literature can be found in oncology journals because oncology is developing very dinamically for obvious reasons. Unfortunately, our paper does not focus strictly on oncological problems. It is an attempt to disscuss the long-term toxic effect of drugs when the exact time of occurrence of the etiological factor is known and the time of occurrence of long-term effects of the therapy is also known. It is the unique phenomenon in medicine. We believe that this will result in new, needed research in the future.

The diagnosis of dental adverse effects is greatly neglected in comparison with the large range of side effects affecting general health. We encounered this when conducting interviews and dental examination in survivors. Patients undergo numerous medical check-ups, and problems occurring in the oral cavity remain unnoticed. Moreover, patients are not aware of the risk of chemotherapy in this regard. Therefore, patients do not associate diagnosed dental abnormalities with anticancer treatment. Consequently, creating a representative cohort of survivors is very difficult. Many parents refuse to have their children undergo dental examination due to greater general health problems.

Few authors have attempted to study similar relationships, but in our opinion they did not adopt appropriate scientifically justified inclusion criteria. This issue was disscussed extensively in the paper. Therefore, publication in an oncology journal seems to be very necessary. Moreover, we believe that it will inspire the oncologists to cooperate in this field.

The authors would like to emphasize that they strongly agree with the Reviewer’s opinion regarding the need to analyze the latest literature. At the same time, they ask for understanding the situation. Below, the authors present the references of 2 recent papers: one on dental abnormalities in general, the other on the topic discussed in the current manuscript. In the first one from 2021, one can find 34 references on dental abnormalities in general, but only 8 of them are published up to 5 years before the manuscript approval for publication. The next paper from 2016 is the latest reference found in the literature on the topic discussed in the current manuscript, which is not publication by the authors. There are 42 references on dental abnormalities in general and only 7 of them are from the last 5 years. Whether the references were relevant and significant for publications was not analyzed in detail.

Your Sincerely,

Dr. Anna Jodłowska

Long-Term Effects of Childhood Cancer Treatment on Dentition and Oral Health: A Dentist Survey Study from the DCCSS LATER 2 Study

Juliette Stolze 1,2,3,* , Kim C. E. Vlaanderen 2, Frederique C. E. D. Holtbach 2, Jop C. Teepen 1 ,

Leontien C. M. Kremer 1,4,5, Jacqueline J. Loonen 6, Eline van Dulmen-den Broeder 7,

Marry M. van den Heuvel-Eibrink 1, Helena J. H. van der Pal 1, Birgitta Versluys 1,

Margriet van der Heiden-van der Loo 1, Marloes Louwerens 8, Judith E. Raber-Durlacher 3,9, Dorine Bresters 1 and Henk S. Brand 2

Cancers 2021, 13, 5264. https://doi.org/10.3390/cancers13215264

  1. O’Leary, M.; Krailo, M.; Anderson, J.R.; Reaman, G.H. Progress in Childhood Cancer: 50 Years of Research. Semin. Oncol. 2008, 35, 484–493. [CrossRef]
  2. Geenen, M.M.; Cardous-Ubbink, M.C.; Kremer, L.C.M.; Bos, C.V.D.; van der Pal, H.J.H.; Heinen, R.C.; Jaspers, M.W.M.; Koning, C.C.E.; Oldenburger, F.; Langeveld, N.E.; et al. Medical assessment of adverse health outcomes in long-term survivors of childhood cancer. J. Am. Med. Assoc. 2007, 297, 2705–2715. [CrossRef] [PubMed]
  3. Oeffinger, K.C.; Mertens, A.C.; Sklar, C.A.; Kawashima, T.; Hudson, M.M.; Meadows, A.T.; Friedman, D.L.; Marina, N.; Hobbie, W.; Kadan-Lottick, N.; et al. Chronic health conditions in adult survivors of childhood cancer. N. Engl. J. Med. 2006, 355, 1572–1582. [CrossRef]
  4. Raber-Durlacher, J.; Epstein, J.; Bresters, D.; Stern-Zivan, L. Oral complications in children with cancer. Eur. J. Paediatr. Dent. Clin. Suppl. 2005, 6, 3–12.
  5. Gawade, P.; Hudson, M.M.; Kaste, S.C.; Neglia, J.P.; Constine, L.S.; Robison, L.L.; Ness, K.K. A systematic review of dental late effects in survivors of childhood cancer. Pediatr. Blood Cancer 2014, 61, 407–416. [CrossRef] [PubMed]
  6. Busenhart, D.M.; Erb, J.; Rigakos, G.; Eliades, T.; Papageorgiou, S.N. Adverse effects of chemotherapy on the teeth and surrounding tissues of children with cancer: A systematic review with meta-analysis. Oral Oncol. 2018, 83, 64–72. [CrossRef]
  7. Seremidi, K.; Kloukos, D.; Polychronopoulou, A.; Kattamis, A.; Kavvadia, K. Late effects of chemo and radiation treatment on dental structures of childhood cancer survivors. A systematic review and meta-analysis. Head Neck 2019, 41, 3422–3433. [CrossRef]
  8. Kaste, S.C.; Goodman, P.; Leisenring, W.; Stovall, M.; Hayashi, R.J.; Yeazel, M.; Beiraghi, S.; Hudson, M.M.; Sklar, C.A.; Robison, L.L.; et al. Impact of Radiation and Chemotherapy on Risk of Dental Abnormalities: A Report from the Childhood Cancer Survivor Study. Cancer Interdiscip. Int. J. Am. Cancer Soc. 2009, 115, 5817–5827. [CrossRef]
  9. Av¸sar, A.; Elli, M.; Darka, Ö.; Pinarli, G. Long-term effects of chemotherapy on caries formation, dental development, and salivary factors in childhood cancer survivors. Oral Surg. Oral Med. Oral Pathol. Oral Radiol. Endodontol. 2007, 104, 781–789. [CrossRef]
  10. Nemeth, O.; Hermann, P.; Kivovics, P.; Garami, M. Long-term effects of chemotherapy on dental status of children cancer survivors. Pediatr. Hematol. Oncol. 2013, 30, 208–215. [CrossRef]
  11. Lauritano, D.; Petruzzi, M. Decayed, missing and filled teeth index and dental anomalies in long-term survivors leukaemic children: A prospective controlled study. Med. Oral Patol. Oral y Cir. Bucal 2012, 17, e977–e980. [CrossRef]
  12. EEffinger, K.E.; Migliorati, C.A.; Hudson, M.M.; McMullen, K.P.; Kaste, S.C.; Ruble, K.; Guilcher, G.M.T.; Shah, A.J.; Castellino, S.M. Oral and dental late effects in survivors of childhood cancer: A Children’s Oncology Group report. Support. Care Cancer 2014, 22, 2009–2019. [CrossRef]
  13. Van Der Pas-Van Voskuilen, I.G.M.; Veerkamp, J.S.J.; Raber-Durlacher, J.E.; Bresters, D.; VanWijk, A.J.; Barasch, A.; McNeal, S.; Gortzak, A.T. Long-term adverse effects of hematopoietic stem cell transplantation on dental development in children. Support Care Cancer 2009, 17, 1169–1175. [CrossRef] [PubMed] Cancers 2021, 13, 5264 12 of 13
  14. Çetiner, D.; Çetiner, S.; Uraz, A.; Alpaslan, G.H.; Alpaslan, C.; Memiko˘ glu, T.U.T.; Karadeniz, C. Oral and dental alterations and growth disruption following chemotherapy in long-term survivors of childhood malignancies. Care Cancer 2018, 27, 1891–1899. [CrossRef] [PubMed]
  15. Quispe, R.A.; Rodrigues, A.C.C.; Buaes, A.M.G.; Capelozza, A.L.A.; Rubira, C.M.F.; Santos, P.S.D.S. A case-control study of dental abnormalities and dental maturity in childhood cancer survivors. Oral Surgery, Oral Med. Oral Pathol. Oral Radiol. 2019, 128, 498–507.e3. [CrossRef] [PubMed]
  16. Wilberg, P.; Kanellopoulos, A.; Ruud, E.; Hjermstad, M.J.; Fosså, S.D.; Herlofson, B.B. Dental abnormalities after chemotherapy in long-term survivors of childhood acute lymphoblastic leukemia 7–40 years after diagnosis. Support. Care Cancer 2016, 24, 1497–1506. [CrossRef]
  17. Goho, C. Chemoradiation therapy: Effect on dental development. Pediatr. Dent. 1993, 15, 6–12.
  18. Sonis, A.; Tarbell, N.; Valachovic, R.W.; Gelber, R.; Schwenn, M.; Sallan, S. Dentofacial development in long-term survivors of acute lymphoblastic leukemia: A comparison of three treatment modalities. Cancer 1990, 66, 2645–2652. [CrossRef]
  19. Kang, C.-M.; Hahn, S.M.; Kim, H.S.; Lyu, C.J.; Lee, J.-H.; Lee, J.; Han, J.W. Clinical risk factors influencing dental developmental disturbances in childhood cancer survivors. Cancer Res. Treat. 2018, 50, 926–935. [CrossRef]
  20. Milgrom, S.A.; van Luijk, P.; Pino, R.; Ronckers, C.M.; Kremer, L.C.; Gidley, P.W.; Grosshans, D.R.; Laskar, S.; Okcu, M.F.; Constine, L.S.; et al. Salivary and Dental Complications in Childhood Cancer Survivors Treated with Radiation Therapy to the Head and Neck: A Pediatric Normal Tissue Effects in the Clinic (PENTEC) Comprehensive Review. Int. J. Radiat. Oncol. 2021, in

press. [CrossRef] [PubMed]

  1. Seremidi, K.; Kavvadia, K.; Kattamis, A.; Polychronopoulou, A. Dental late effects of antineoplastic treatment on childhood cancer survivors: Radiographic findings. Int. J. Paediatr. Dent. 2021, in press. [CrossRef]
  2. Maguire, A.; Craft, A.W.; Evans, R.G.B.; Amineddine, H.; Kernahan, J.; MacLeod, R.I.; Murray, J.J.;Welbury, R. The long-term effects of treatment on the dental condition of children surviving malignant disease. Cancer 1987, 60, 2570–2575. [CrossRef]
  3. Macleod, R.I.; Welbury, R.R.; Soames, J.V. Effects of cytotoxic chemotherapy on dental development. J. R. Soc. Med. 1987, 80, 207–209. [CrossRef] [PubMed]
  4. Tanaka, M.; Kamata, T.; Yanagisawa, R.; Morita, D.; Saito, S.; Sakashita, K.; Shiohara, M.; Kurita, H.; Koike, K.; Nakazawa, Y. Increasing Risk of Disturbed Root Development in Permanent Teeth in Childhood Cancer Survivors Undergoing Cancer Treatment at Older Age. J. Pediatr. Hematol. 2017, 39, e150–e154. [CrossRef] [PubMed]
  5. Ritwik, P. Dental care for patients with childhood cancers. Ochsner J. 2018, 18, 351–357. [CrossRef] [PubMed]
  6. Hölttä, P.; Hovi, L.; Saarinen-Pihkala, U.M.; Peltola, J.; Alaluusua, S. Disturbed root development of permanent teeth after pediatric stem cell transplantation. Dental root development after SCT. Cancer 2005, 103, 1484–1493. [CrossRef]
  7. Knol, M.J.; Le Cessie, S.; Algra, A.; Vandenbroucke, J.P.; Groenwold, R.H. Overestimation of risk ratios by odds ratios in trials and cohort studies: Alternatives to logistic regression. Can. Med. Assoc. J. 2012, 184, 895–899. [CrossRef]
  8. Green, D.M.; Nolan, V.G.; Ms, P.J.G.; Ms, J.A.W.; Srivastava, D.; ScD, W.M.L.; Neglia, J.; Sklar, C.A.; Kaste, S.C.; Hudson, M.M.; et al. The cyclophosphamide equivalent dose as an approach for quantifying alkylating agent exposure: A report from the Childhood Cancer Survivor Study. Pediatr. Blood Cancer 2014, 61, 53–67. [CrossRef]
  9. Cubukcu, C.E.; Sevinir, B.; Ercan, I. Disturbed dental development of permanent teeth in children with solid tumors and lymphomas. Pediatr. Blood Cancer 2012, 58, 80–84. [CrossRef]
  10. Apajalahti, S.; Hölttä, P.; Turtola, L.; Pirinen, S. Prevalence of short-root anomaly in healthy young adults. Acta Odontol. Scand. 2002, 60, 56–59. [CrossRef]
  11. Polder, B.J.; Hof, M.A.V.; Van Der Linden, F.P.G.M.; Kuijpers-Jagtman, A.M. A meta-analysis of the prevalence of dental agenesis of permanent teeth. Community Dent. Oral Epidemiol. 2004, 32, 217–226. [CrossRef] [PubMed]
  12. Ooshima, T.; Ishida, R.; Mishima, K.; Sobue, S. The prevalence of developmental anomalies of teeth and their association with tooth size in the primary and permanent dentitions of 1650 Japanese children. Int. J. Paediatr. Dent. 2009, 6, 87–94. [CrossRef] [PubMed]
  13. Carrillo, C.M.; Corrêa, F.N.P.; Lopes, N.N.F.; Fava, M.; Filho, V.O. Dental anomalies in children submitted to antineoplastic therapy. Clinics 2014, 69, 433–437. [CrossRef]
  14. Minicucci, E.M.; Lopes, L.F.; Crocci, A.J. Dental abnormalities in children after chemotherapy treatment for acute lymphoid leukemia. Leuk. Res. 2003, 27, 45–50. [CrossRef]
  15. Dahllöf, G.; Jönsson, A.; Ulmner, M.; Huggare, J. Orthodontic treatment in long-term survivors after pediatric bone marrow transplantation. Am. J. Orthod. Dentofac. Orthop. 2001, 120, 459–465. [CrossRef]
  16. Roscoe, M.G.; Meira, J.; Cattaneo, P.M. Association of orthodontic force system and root resorption: A systematic review. Am. J. Orthod. Dentofac. Orthop. 2015, 147, 610–626. [CrossRef]
  17. Pajari, U.; Lanning, M. Developmental defects of teeth in survivors of childhood ALL are related to the therapy and age at diagnosis. Med Pediatr. Oncol. 1995, 24, 310–314. [CrossRef]
  18. Koch, G.; Thesleff, I.; Kreiborg, S. Tooth Development and Disturbances in Number and Shape of Teeth. In Pediatric Dentistry—A Clinical Approach; Koch, G., Poulsen, S., Eds.;Wiley: Oxford, UK, 2009; pp. 183–196.
  19. Koch, G.; Kreiborg, S.; Andreasen, J.O. Eruption and Shedding of Teeth. In Pediatric Dentistry—A Clinical Approach, 3rd ed.; Wiley: Oxford, UK, 2016; pp. 40–54. Cancers 2021, 13, 5264 13 of 13
  20. Németh, O. Dental and Craniofacial Effects on Childhood Cancer Survivors. In Pediatric Cancer Survivors; Wonders, K., Stout, B., Eds.; IntechOpen: Rijeka, Croatia, 2017.
  21. Hölttä, P.; Alaluusua, S.; Saarinen-Pihkala, U.M.; Wolf, J.; Nyström, M.; Hovi, L. Long-term adverse effects on dentition in children with poor-risk neuroblastoma treated with high-dose chemotherapy and autologous stem cell transplantation with or without total body irradiation. Bone Marrow Transplant. 2002, 29, 121–127. [CrossRef]
  22. Hölttä, P.; Alaluusua, S.; Saarinen-Pihkala, U.M.; Peltola, J.; Hovi, L. Agenesis and microdontia of permanent teeth as late adverse effects after stem cell transplantation in young children. Cancer 2004, 103, 181–190. [CrossRef]
  23. Maciel, J.C.C.; de Castro, C.G.; Brunetto, A.L.; Di Leone, L.P.; da Silveira, H.E.D. Oral health and dental anomalies in patients treated for leukemia in childhood and adolescence. Pediatr. Blood Cancer 2009, 53, 361–365. [CrossRef]
  24. Duggal, M. Root surface areas in long-term survivors of childhood cancer. Oral Oncol. 2003, 39, 178–183. [CrossRef]
  25. Hsieh, S.G.-S.; Hibbert, S.; Shaw, P.; Ahern, V.; Arora, M. Association of cyclophosphamide use with dental developmental defects and salivary gland dysfunction in recipients of childhood antineoplastic therapy. Cancer 2011, 117, 2219–2227. [CrossRef]
  26. Teepen, J.C.; Van Leeuwen, F.E.; Tissing, W.J.; Broeder, E.V.D.-D.; Heuvel-Eibrink, M.M.V.D.; Van Der Pal, H.J.; Loonen, J.; Bresters, D.; Versluys, B.; Neggers, S.J.C.M.M.; et al. Long-Term Risk of Subsequent Malignant Neoplasms after Treatment of Childhood Cancer in the DCOG LATER Study Cohort: Role of Chemotherapy. J. Clin. Oncol. 2017, 35, 2288–2298. [CrossRef]

Antineoplastic chemotherapy and congenital tooth abnormalities in children and adolescents

Ewa Krasuska-Sławińska1, Agnieszka Brożyna2, Bożenna Dembowska-Bagińska2, Dorota Olczak-Kowalczyk3

Contemp Oncol (Pozn) 2016; 20 (5): 394–401

DOI: 10.5114/wo.2016.64602

  1. Stiller. Epidemiology and genetics of childhood cancer. Oncogene 2004; 23: 6429-44.
  2. SEER Cancer Statistics review 1975-2010. Rieg LAG, et al. (eds.). National Cancer Institute Nethesda MD, 2013. 400 contemporary oncology
  3. Kurt BA, Armstrong GT, Cash DK, Krasin MJ, Morris EB, Spunt SL, Robison LL, Hudson MM. Primary care management of the childhood cancer survivor. J Pediatr 2008; 158: 458-66.
  4. Miller JL, McLeod HL. Pharmacogenesis of cancer chemotherapy- induced toxicity. J Support Oncol 2007; 5: 9-14.
  5. Huszno J, Nowara E. Pharmacocinetics and pharmacogenetics in breast cancer patients systemic treatment. Onkol Prakt Klin 2010; 6: 159-70.
  6. Pawlicki M, Wiczyńska B. New anticancer drugs – future directions. Nowotwory J Oncol 2001; 51: 507-14 (Polish).
  7. Olczak-Kowalczyk D, Perek D, Daszkiewicz M, Adamowicz-Klepalska B, Dembowska-Bagińska B, Daszkiewicz P. Oral pathology in children harboring neoplastic diseases. Own experience. Nowa

Stomatol 2003; 4: 175-9 (Polish).

  1. Ferlay J, et al. Globocan 2002: Cancer incidence mortality and prevalence worldwide. IARC cancer Nase No 5 Version 2,0. IARC Press, Lyon 2004.
  2. Teuffel O, Dettling M, Cario G, Stanulla M, Schrappe M, Buhlmann P, Niggli FK, Schafer BW. Gene expression profiles and risk stratification in childhood acute lymphoblastic leukemia. Haematologica 2004; 89: 801-8.
  3. Sonis AL, Waber DP, Sallan S, Trabell NJ. The oral health of long term survivors of acute lymphoblastic leukemia, a comparison of tree treatment modalities. Eur J Cancer B Oral Oncol 1995; 31B: 250-252.
  4. Hutton A. The oral health needs children, adolescents and young adults after cancer therapy for solid tumor. University of Brimingham 2008. Available at: http://etheses.bham.ac.uk/142/1/Hutton08MPhil. pdf
  5. Pajari U, Yliniemi R. The risk of dental caries in childhood cancer is not high in the teeth are caries-free at diagnosis. Pediatr Hematol Oncol 2001; 18: 181-5.
  6. Macleod RI, Welbury RR, Soames JV. Effects of cytotoxic chemotherapy on dental development. J R Soc Med 1987; 80: 207-9.
  7. Maciel JC, de Castro CG Jr, Brunetto AL, Di Leone LP, da Silveira HE. Oral health and dental anomalies in patients treated for leukemia in childhood and adolescence. Pediatr Blood Cancer 2009; 53: 361-5.
  8. Sosnowska-Boroszko A, Gordon A, Siemińska J, et al. Enamel abnormalities in permanent dentition of school children. Nowa Stomatol 2002; 3: 116-21 (Polish).
  9. Robinson C, Kirkham J, Brookes SJ, Shore R. Chemistry of mature enamel. In: Dental enamel – formation to destruction. Kirkham RC (eds.). J CRC Press 1995; 167-188.
  10. Alpaslan G, Alpaslan C, Gogen H, Oğuz A, Cetiner S, Karadeniz C. Disturbances in oral dental structures in patients with pediatric lymphoma after chemotherapy. Oral Surg Oral Med Oral Patol Oral Radiol Endod 1999; 87: 317-321.
  11. Olender E, Kamińska A, Urynowska-Tyszkiewicz I, Wanyura H. Histological aspects and the molecular controlling mechanism of the natural dental development. Czas Stomatol 2010; 63: 543-50

(Polish).

  1. Pajari U, Lanning M, Larmas M. Prevalence and location of enamel opacities in children after anti-neoplastic therapy. Comm Dent Oral Epidemiol 1988; 16: 222-6.
  2. Holtta P, Alaluusua S, Saarinen-Pihkala UM, Wolf J, Nystrom M, Hovi L. Long-term adverse effects on dentition in children with poor-risk neuroblastoma treated with high-dose chemotherapy

and autologous stem cell transplantation with or without total body irradiation. Bone Marrow Transpl 2002; 29: 121-7.

  1. Holtta P, Alaluusua S, Saarinen-Pihkala UM, Wolf J, Nystrom M, Hovi L. Agenesis and microdontia of permanent teeth as late adverse effects after stem cell transplantation in young children. Cancer 2005; 103: 181-91.
  2. Holtta P. Developmental dental defects in children who reside by river polluted by dioxins and furans. Arch Environ Health 2001; 56: 522-8.
  3. Holtta P. Developmental aberrations of permanent teeth after high-dose anticancer therapy in childhood. University of Helsinki 2005. Available at: http://ethesis.helsinki.fi/julkaisut/laa/hamma/

vk/holtta/developm.pdf.

  1. Jones TE, Henderson JS, Jahnson RB. Effects of doxorubicin on human dental pulp cells in vitro. Cell Biol Toxicol 2005; 21: 207-14.
  2. Hsieh SG, Hibbert DS, Sha P, Ahern V, Arora M. Association of cyclophosphamide use with dental defects and salivary gland dysfunction in recipients of childhood antineoplastic therapy. Cancer 2011; 117: 2219-27.
  3. Lyaruu DM, van Duin MA, Bervoets TJ, Woltgens JH, Bronckers AL. Effects of actinomycin D developing hamster molar tooth germs in vitro. Eur J Oral Sci 1997; 105: 52-8.
  4. Marec-Berard P, Chaux-Bodard AG, Langrange H, Gourmet R, Bergeron C. Long-term effects of chemotherapy on dental status in children treated for nephroblastoma. Pediatric Hematol Oncol

2005; 22: 581-8.

  1. Minicucci EM, Lopez LF, Crocci AJ. Dental abnormalities in children after chemotherapy treatment for acute leukemia. Leukemia Research 2003; 25: 45-50.
  2. Jaffe N, Toth BB, Hoar RE, Ried HL, Sullivan MP, McNeese MD. Dental and maxillofacial abnormalities in long-term survivors of childhood cancer: effects of treatment with chemotherapy and radiation to the head and neck. Pediatr 1984; 73: 816-23.
  3. Remmers D, Bokkerink JP, Katsaros C. Microdontia after chemotherapy in a child treated for neuroblastoma. Orthod Craniofacial Res 2006; 9: 206-10.
  4. Rosenberg S, Kolodney J, Wong GY, Murphy L. Altered dental root development in long-term survivors of pediatric acute lymphoblastic leukemia. Cancer 1987; 59: 1640-8.
  5. Najafi SH, Tohidastakrad Z, Momenbeitollahi J. The long-term effects of chemo radiotherapy on oral health and dental development in childhood cancer. J Dent (Teheran) 2011; 8: 39-43.
  6. Hwang SY, Yoon RK. Developmental dental defects linked with chemotherapy: a case report. J Clin Pediatr Dent 2011; 35: 309-13.
  7. Maguire A, Welbury RR. Long-term effects of antineoplastic chemotherapy and radiotherapy on dental development. Dent Update 1996; 23: 188-94.
  8. Ilgenili T, Oren H, Uysal K. The acute effects of chemotherapy upon the oral cavity: Prevention and management. Turk J Cancer 2001; 31: 93-105.
  9. Oğuz A, Cetiner S, Karadeniz C, Alpaslan G, Alpaslan C, Pinarli G. Long-term effects of chemotherapy on orodental structures in children with non-Hodkin’s Lymphoma. Eur J Oral Sci 2004; 112: 8-11.
  10. Sonis AL, Tarbell N, Valachovic RW, Gelber R, Schewnn m, Sallam s. Dentofacial development in long term survivors of acute lymphoblastic leukemia. Cancer 1990; 60: 2645-52.
  11. Kaste SC, Hopkins KP, Jones D, Crom D, Greenwald CA, Santana VM. Dental abnormalities im children treated for acute lymphoblastic leukemia. Leuk 1997; 11: 792-6.
  12. Lukinmaa PL, Sahlberg C, Leppaniemi A, Partanen AM, Kovero O, Pohjanvirta R, Tuomisto J, Alaluusua S. Arrest of rat molar tooth development by lactational exposure to 2,3,7,8-tetrachlorodibenzo- p-dioxin. Toxicol App Pharmacol 2001; 173: 38-47.
  13. Nieminen P, Arte S, Tanner D, Paulin L, Alaluusua S, Thesleff I, Pirinen S. Identification of a nonsense mutation in the PAX9 gene in molar oligodontia. Eur J Hum Genetic 2001; 9: 743-6.
  14. Partanen AM, Kiukkonen A, Sahlberg C, Alaluusua S, Thesleff I, Pohjanvirta R, Lukinmaa PL. Developmental toxicity of dioxin to mouse embryonic teeth in vitro: arrest of tooth morphogenesis involves stimulation of apoptotic program in the dental epithelium. Toxicol App Pharmacol 2004; 194: 24-33.
  15. Lammi L, Arte S, Somer M, Jarvinen H, Lahermo P, Thesleff I, Pirinen S, Nieminen P. Mutation in AXIN2 course familial tooth agenesis and predispose to colorectal cancer. Am J Hum Genet 2004; 74: 1043-50.
  16. Boyle P, Ferlay A. Cancer incidence and mortality in Europe. Ann Oncol 2004; 16: 481-8.
  17. Stachowicz-Stencel T, Stefanowicz J, Bień E, Balcerska A. Long-term effects of the treatment malignant neoplasm in children. Forum Med Rodz 2009; 3: 485-93 (Polish).
  18. Jemal A, Siegeil R, Ward E, Muray T, Xu J, Thun MJ. Cancer statistics. Cancer J Clin 2007; 57: 43-66.
  19. Liu YJ, Zhu GP, Guan XY. Comparison of the NCI-CTCAE version 4.0 and version 3.0 in assessing chemoradiation-induced oral mucositis. Antineoplastic chemotherapy and congenital tooth abnormalities in children and adolescents 401 tis for locally advanced nasopharyngeal carcinoma. Oral Oncol 2012; 48: 554-9.
  20. A review of the developmental defects of enamel index (DDE Index). Commission on Oral Health, Research & Epidemiology. Report of an FDI Working Group. Int Dent J 1992; 42: 411-26.
  21. Nasman M, Forsberg CM, Dahllof G. Long-term dental development in children after treatment for malignant disease. Eur J Orthod Sci 1997; 19: 151-9.
  22. Dahllof G, Nasman M. Effect of chemotherapy of dental maturity in children with hematological malignancies. Pediatr Dent 1989; 11: 303-6.
  23. Lauritano D, Petruzzi M. Decayed, missing and filled teeth index and dental abnormalities in long-term survivors leukemic children: a prospective controlled study. Med Oral Patol Cir Bucal 2012; 17: e977-80.
  24. Al-Ghani BA, Hasan NM, Hassan JM. Enamel hypoplasia in patients with acute lymphoblastic leukemia. J Bagh Colleg Dent 2005; 17: 53-6.
  25. Dodan C, Haytac C, Antem B, Babmaz Y, Tanyely A. Oral health in children with acute lymphoblastic leukemia and lymphoma. Turk J Hematol 2001; 18: 179-83.
  26. Avşar A, Elli M, Darka O, Pinarli G. Long-term effects of chemotherapy on caries formation, dental development, and salivary factors in childhood cancer survivors. Oral Surg Oral Med Oral Pathol Oral Radiol Endod 2007; 104: 781-9.
  27. Greaves P. Mouth and oropharynx. Greaves P (ed.). Histopathology of preclinical toxicity studies. AP: 2012; 325-332.
  28. Nunn JH, Welbury RR, Gordon PH, Kernahan J, Craft AW. Dental caries and dental anomalies in children treated by chemotherapy for malignant disease: a study in the north of England. Int J Paediatr Dent 1991; 1: 131-5.
  29. Kinirons MJ, Fleming P, Boyd D. Dental caries experience of children in remission from acute lymphoblastic leukemia in relation to the duration of treatment and the period of time in remission. Int J Paediatr Dent 1995; 5: 169-72.
  30. Cubucku CE, Sevinir B, Ercan I. Disturbed dental development of permanent teeth in children with solid tumors and lymphomas. Pediatr Blood Cancer 2012; 58: 80-4.
  31. Nishimura S, Inada H, Sawa Y, Ishikawa H. Risk factors to cause of tooth anomalies in chemotherapy of pediatric cancers. Eur J Cancer Care 2013; 22: 353-60.
  32. Dembowska-Bagińska B. Health status and psychosocial problems of children and adolescents after treatment of malignant childhood central nervous system tumors. The Children’s Memorial Health Institute Warsaw, 2008 (Polish).
  33. Dembowska-Bagińska B. Long-term effect of multimodal treatment of childhood malignancies. Stand Med 2001; 3: 45-51 (Polish).
  34. Olczak-Kowalczyk D, Daszkiewicz M, Adamowicz-Klepalska B, Mielnik-Błaszczyk M, Dembowska-Bagińska B, Perek D. The status of dentition and oral hygiene in children after anticancer treatment. Ann Acad Med 2004; 34: 237-55 (Polish).
  35. Woltgens JH, Lyaruu DM, Bronckers AL, Duin MA. Effects of methotrexate on cell proliferation in developing hamster tooth germs in vitro. Eur J Oral Sci 1998; 106: 156-9.

Reviewer 2 Report

Comments and Suggestions for Authors

It is important to acknowledge the remarkable enhancement in the quality of the article made by the authors. The implemented improvements are remarkable and have rendered the content highly suitable for publication. The revisions and adjustments carried out reflect meticulous and in-depth work, resulting in a substantially improved article, now ready for acceptance for publication.

Author Response

Anna Jodłowska, PhD

Department of Pediatric Dentistry, Medical University of Silesia in Katowice, Poland, 41-800 Zabrze, Pl. Traugutta 2

Lidia Postek-Stefańska, PhD with habilitation

Department of Pediatric Dentistry, Medical University of Silesia in Katowice, Poland, 41-800 Zabrze, Pl. Traugutta 2

November 12, 2023

Refers to: Manuscript ID: cancers-2628352
Type of manuscript: Article
Title: "Tooth abnormalities and their age dependent occurrence in leukemia survivors"
Special Issue: "Side Effects of Anticancer Therapy: Prevention and Management"

Dear Reviewer,

The authors kindly thank you for acceptance of our last version of manuscript and understanding the reasons why certain conditions cannot be met. Once again we would like to thank you for invaluable contribution to the creation of the manuscript. We are grateful that you appreciated our efforts in working on the manuscript and you recommended it for publication.

Your Sincerely,

Dr. Anna Jodłowska
